**Article** https://doi.org/10.1038/s41467-023-40473-w

# Extending the coherence of spin defects in hBN enables advanced qubit control and quantum sensing

Roberto Rizzato [1,2,9] ✉, Martin Schalk[3,4,9], Stephan Mohr[1], Jens C. Hermann[1,4], Joachim P. Leibold[1,5], Fleming Bruckmaier[1], Giovanna Salvitti [1,6], Chenjiang Qian [3], Peirui Ji[3], Georgy V. Astakhov [7], Ulrich Kentsch[7], Manfred Helm[7,8], Andreas V. Stier [3,4], Jonathan J. Finley [3,4] & Dominik B. Bucher [1,4] ✉

Negatively-charged boron vacancy centers ($V_B^-$) in hexagonal Boron Nitride (hBN) are attracting increasing interest since they represent optically-addressable qubits in a van der Waals material. In particular, these spin defects have shown promise as sensors for temperature, pressure, and static magnetic fields. However, their short spin coherence time limits their scope for quantum technology. Here, we apply dynamical decoupling techniques to suppress magnetic noise and extend the spin coherence time by two orders of magnitude, approaching the fundamental $T_1$ relaxation limit. Based on this improvement, we demonstrate advanced spin control and a set of quantum sensing protocols to detect radiofrequency signals with sub-Hz resolution. The corresponding sensitivity is benchmarked against that of state-of-the-art NV-diamond quantum sensors. This work lays the foundation for nanoscale sensing using spin defects in an exfoliable material and opens a promising path to quantum sensors and quantum networks integrated into ultra-thin structures.

Optically addressable spin defects in semiconductors are promising systems for various applications in quantum science and technology, including sensing and metrology[1–4]. In contrast to other defects typically hosted in 3D crystals[5], the recently discovered boron vacancy center ($V_B^-$) in hexagonal boron nitride (hBN)[6,7] is embedded in a van der Waals material which can be exfoliated down to the limit of a single monolayer[8,9] (Fig. 1a). Such a unique feature would be advantageous for a wide range of applications where a minimal spatial separation of the spin defect to a specific target is highly desired. For example, in nanoscale quantum sensing, spatial resolution is determined by the proximity of the defect to the test object[10–12], or for integrated quantum photonic devices van der Waals materials can be readily exfoliated onto different substrates and used as spin-photon interfaces[4]. Furthermore, the $V_B^-$ center, incorporated in ultra-thin hBN foils, allows for easier integration of manipulable qubits in 2D heterostructures. This possibility opens up unexplored paths for investigating novel composite materials and phenomena in nanoelectronics, nanophotonics, and spintronics[8,9,13–15]. The first protocols for generating $V_B^-$ centers in hBN have been recently presented[16,17], and their spectroscopic characterization has been accomplished in several studies[7,18–21].

[1]Technical University of Munich, TUM School of Natural Sciences, Department of Chemistry, Lichtenbergstraße 4, Garching bei München 85748, Germany. [2]University of Bari, Department of Physics "M. Merlin", Via Amendola 173, Bari 70125, Italy. [3]Walter Schottky Institute, TUM School of Natural Sciences, Am Coulombwall 4, Garching bei München 85748, Germany. [4]Munich Center for Quantum Science and Technology (MCQST), Schellingstr. 4, München D-80799, Germany. [5]Technical University of Munich, TUM School of Natural Sciences, Department of Physics, James-Franck-Str. 1, Garching bei München 85748, Germany. [6]University of Bologna, Department of Chemistry "G. Ciamician", Via Selmi, 2, Bologna 40126, Italy. [7]Helmholtz-Zentrum Dresden-Rossendorf, Institute of Ion Beam Physics and Materials Research, Bautzner Landstraße 400, Dresden 01328, Germany. [8]TU Dresden, 01062 Dresden, Germany. [9]These authors contributed equally: Roberto Rizzato and Martin Schalk. ✉e-mail: roberto.rizzato@tum.de; dominik.bucher@tum.de

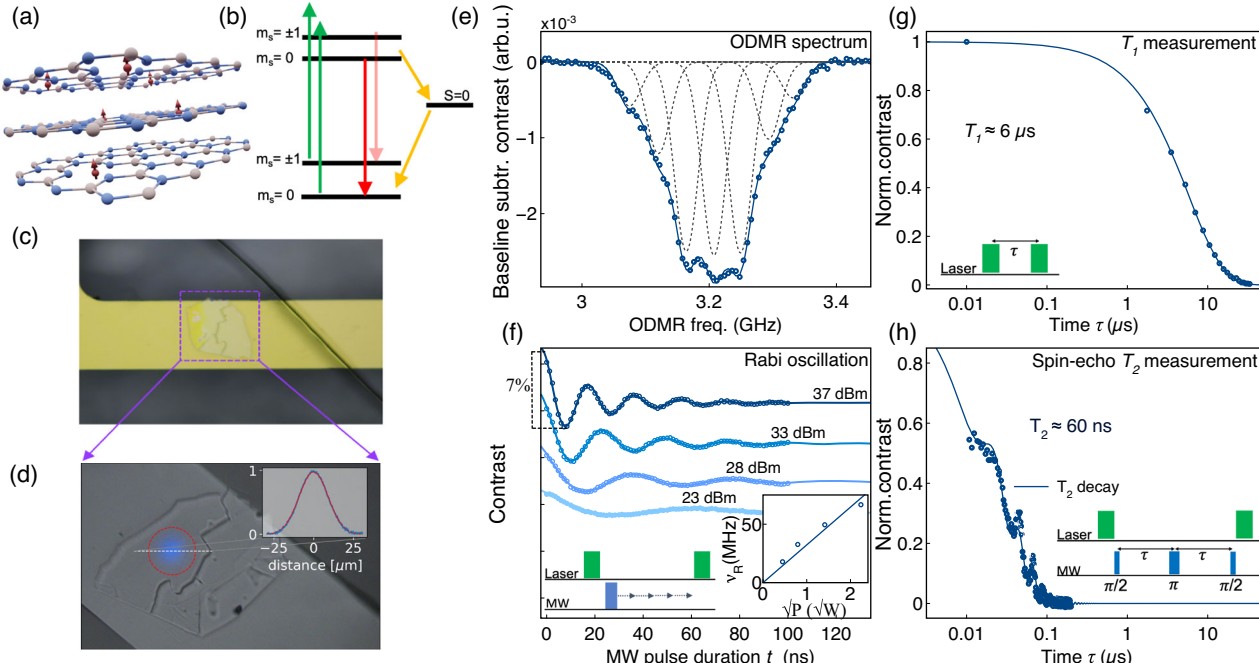

**Fig. 1 | Characterization of the $V_B^-$ defect's spin properties. a** Idealized structure of $V_B^-$ spin defects (red arrows) in hBN layers. Nitrogen atoms are displayed in blue, boron in gray. The number of spin defects is only for illustrative purpose and does not reflect the actual density. **b** Simplified energy levels displaying the ground and excited triplet states, the optical excitation transitions (green arrows), the fluorescence pathways (red arrows), and the non-radiative pathways through intersystem crossing (ISC) to the $S = 0$ singlet state (yellow arrows) which allows for spin-state dependent optical readout. **c** Microscope image of the hBN flake positioned on the gold microwave (MW) stripline. **d** Image of the sample from the optically detected magnetic resonance (ODMR) setup with the laser spot (blue, false color) for the initialization/interrogation of the spin ensembles. In the inset, the Gaussian profile of the laser spot is shown. **e** ODMR spectrum of the $|0\rangle \rightarrow |-1\rangle$ transition at -8 mT. The data points are fitted with single Gaussian functions (dotted lines) that are summed up to give the overall spectral lineshape (solid-blue line). **f** Rabi oscillations for four different MW powers. The MW frequency was set at the center of the ODMR spectrum in **e**. Inset, left: pulse sequence for the Rabi experiment. Right: dependence of the Rabi frequency $\nu_R$ versus the square root of the microwave power $P$. **g** Semi-log plot of the spin-lattice relaxation time decay ($T_1$) measured by the recovery pulse sequence (inset). A $T_1$ time of -6 μs is extracted from the mono-exponential fit. **h** Semi-log plot of the coherence time $T_2$ measured with the depicted spin-echo sequence. $T_2 = 60$ ns is obtained from a stretched exponential fit. A strong signal modulation is superimposed to the same $T_2$ decay curve with a frequency of -45 MHz.

Importantly, by detecting the changes in their optically detected magnetic resonance (ODMR) spectra, the $V_B^-$ centers demonstrated to work as sensors for temperature, pressure, and static magnetic fields, in some cases being competitive with more mature spin-defect-based sensors[22]. Based on these results, first applications have been demonstrated for the magnetic and temperature nanoscale imaging of low-dimensional materials[23,24]. Coherent control of ensembles of $V_B^-$ centers has recently been shown[18,19,25-27], although coherence times ≲100 nanoseconds have been reported[27-29]. These short timescales significantly restrict the utility of such spin qubits and discourage the development of applications based on coherent spin manipulation.

In this work, we combine efficient microwave (MW) delivery and precise MW control to perform dynamical decoupling schemes, such as Carr-Purcell-Meiboom-Gill[30] (CPMG) to efficiently suppress magnetic noise from the spin bath and increase the $V_B^-$ coherence. We extend the room-temperature $T_2$ coherence time of $V_B^-$ ensembles in hBN by nearly two orders of magnitude. Furthermore, we generate $V_B^-$ dressed-states (DS) by applying spinlock pulses and show that spin coherence can be preserved for a time ($T_{1\rho}$), which is on the same order as the spin-lattice relaxation time $T_1$. These results show consistency with the recent findings by Ramsay et al.[29], who were able to increase the coherence time to a similar extent using an alternative approach based on a continuous concatenated dynamic decoupling (CCD) method[31-33]. In addition, our results offer further evidence that enhanced coherence enables the detection of radiofrequency (RF) signals, even with a frequency resolution that is far beyond the intrinsic coherence time of the spin defect. Furthermore, we investigate the potential of these systems by experimentally assessing their sensitivity to RF fields. With a particular focus on exploring their possible applications in nanoscale spin sensing, we compare their performance to that of typical spin-defect-based sensors relying on NV centers in diamond. This work broadens the applicability of $V_B^-$ defects in hBN, opening up new opportunities for nanoscale quantum sensing and technology.

## Results and discussion

### Characterization of $V_B^-$ spin properties

The negatively charged boron vacancy center ($V_B^-$) consists of a missing boron atom in the hBN lattice surrounded by three equivalent nitrogen atoms (Fig. 1a). Ten electrons occupying six defect orbitals result in a triplet $S = 1$ ground state consisting of the $|m_s = 0\rangle$ ($|0\rangle$) and $|m_s = \pm 1\rangle$ ($|\pm 1\rangle$) spin states[6,34]. At zero magnetic field, the $|\pm 1\rangle$ states are degenerate but separated in energy from the $|0\rangle$ state due to a zero-field splitting (ZFS) of $D \sim 3.47$ GHz. The transition from the ground state to the excited state by green laser illumination (e.g., $\lambda = 532$ nm) is followed by a phonon-assisted radiative decay with broad photoluminescence (PL) peaking at -850 nm[35,36]. A spin-state-dependent relaxation path through inter-system crossing (ISC) leads to two important consequences: (1) the $V_B^-$ defects can be optically initialized into the $|0\rangle$ state under ambient conditions; (2) the $|0\rangle$ and $|\pm 1\rangle$ states can be distinguished by their spin-state dependent PL (Fig. 1b)[7,19,34,37].

All experiments presented in this work were conducted under ambient conditions on $V_B^-$ ensembles obtained by He+ implantation of hBN flakes (-100 nm-thick, details in the Methods section). As depicted schematically in Fig. 1c, the flakes are directly transferred onto a gold

MW microstripline that is used for $V_B^-$ spin manipulation. A microscope for spatially resolved ODMR measurements has been built that can address defined areas of the sample with a laser spot size of ~20 μm diameter (see Fig. 1d and Experimental setup in Methods).

As a first characterization experiment, we measured the electron-spin resonance (ESR) spectrum by performing ODMR at a bias magnetic field $B_0$ ~ 8 mT. The spectrum shows a broad resonance corresponding to the $|0\rangle \rightarrow |-1\rangle$ transition (Fig. 1e), revealing characteristic features due to the strong hyperfine (HF) coupling of the $V_B^-$ electronic spin with the three equivalent $^{14}$N nuclei. The HF lines can be fitted with seven Gaussian functions, separated by ~ 44 MHz[7,18,26,27]. Their lineshape indicates the inhomogeneously broadened nature of the spectrum, where the electronic spins experience a broad distribution of local magnetic fields due to the intricate HF structure. Based on this initial observation, we anticipate that the coherence properties of the $V_B^-$ electronic spins will be significantly impacted by hyperfine interactions with nearby nuclear spins. To perform coherent control of the $V_B^-$ centers, we run Rabi experiments with the MW frequency at the central peak of the ODMR spectrum and monitor the fluorescence contrast while sweeping the MW-pulse duration $t_p$. Figure 1f shows Rabi oscillations for different MW amplitudes. We observe a 6–7% fluorescence contrast at maximum amplitude with a π-pulse duration of $t_p^{(\pi)} = 7.5$ ns, corresponding to a Rabi frequency $\nu_R$ ~67 MHz. Maximizing the Rabi frequency is crucial for two reasons: (1) it allows for short pulses that are necessary for an efficient spin manipulation, especially in the presence of fast spin-dephasing, (2) it allows us to drive a large bandwidth of the ODMR spectrum. Fourier transformation of our rectangular π/2-pulses gives an excitation bandwidth of approximately $1/t_p^{(\pi/2)} = 250$ MHz for a ~ 4 ns duration. This has the positive effect of increasing the observable contrast whilst reducing detrimental spectral diffusion effects.

To characterize the $V_B^-$ relaxation properties, we measure the spin-lattice relaxation time $T_1$ using a protocol consisting of two 5 μs-long laser pulses for initialization and readout separated by the sweep time $\tau$. A time constant $T_1$ ~ 6 μs is extracted from a mono-exponential fit of the resulting contrast decay (see Fig. 1g). This value is in agreement with a previous work[38], where the same conditions for defects generation in hBN were utilized (refer to Sample preparation in the Methods section). However, other groups have reported 2–3 times longer $T_1$ values[18,19] which is likely due to differences in the sample preparation.

Furthermore, we measure the native coherence time $T_2$ using the spin-echo sequence depicted in Fig. 1h. Here, the fluorescence contrast is detected while sweeping the free precession time $\tau$. Fitting the resulting signal decay with a stretched exponential function[39,40] (see pulse sequences, normalizations and fittings in Methods), gives a time constant $T_2$ ~60 ns, which is similar to reported values[27,28]. Such a coherence time is short, especially compared to other ensembles of spin defects in 3D-host materials, where typical $T_2$ times lie in the vicinity of microsecond timescales at room-temperature[11]. Interestingly, a clear oscillation appears superimposed on the spin-echo decay. This modulation is tentatively assigned to the interaction of the $V_B^-$ electronic spin with the three $^{14}$N nuclei since the frequency of ~ 45 MHz, fit by a cosine function (see Methods and Supplementary Note 1), matches the HF coupling[7]. An improvement of the coherence time is a critical step for utilizing $V_B^-$ in hBN in advanced quantum technologies, where qubit coherent manipulation and quantum information storage/retrieval are each essential preconditions.

## Extension of the $V_B^-$ coherence

Dynamical decoupling (DD) techniques are traditional tools of nuclear magnetic resonance spectroscopy (NMR)[30,41] and have been extensively applied in the past years to prolong the coherence of spin defects in solid-state materials[11,42–47]. Here, we apply this approach to $V_B^-$ ensembles in hBN to improve their short coherence times and unlock possibilities based on advanced spin manipulation.

A significant source of $V_B^-$ decoherence is likely to be found in spin 'flip-flops' from the bath spins (nuclei or paramagnetic impurities) surrounding the defects[27,28]. These processes cause random magnetic field fluctuations felt by the electronic spin on a time scale set by the average interactions involved[39,45,48,49]. We show that the CPMG DD protocol[41] can be applied to decouple the $V_B^-$ spins from magnetic noise. This is done by applying resonant MW π-pulses, following the scheme $(\pi/2)_y[-\tau - (\pi)_x - \tau]_N$ (see bottom-left inset in Fig. 2a), that have the effect to periodically re-phase the $V_B^-$ superpositions and sustain their coherence over longer timescales. Then, the coherence is mapped to the spin populations via a last $(\pi/2)_y$-pulse for the final optical readout. If $\tau$ is shorter than the spin bath fluctuation correlation time $t_c$, the magnetic noise appears to be time-independent[39,48], and the train of π-pulses can effectively cancel it out.

In Fig. 2a, we show the signal intensity of the spin-echo obtained after $N$ MW π-pulses upon increasing the delay $\tau$ between them. We plot the decays against the total pulse sequence time $t_s = 2N\tau$. This results in multiple decay curves, showing how fast decoherence occurs depending on the number of π-pulses utilized for noise suppression. Then, we extract the characteristic $T_2^{(N)}$ times by fitting every curve (see Methods). We observe a factor ~ 70 increase in the coherence time ($T_2^{(1000)} \sim 4.2\,\mu s$) by applying up to 1000 π-pulses, with respect to the spin-echo with a single π-pulse ($T_2$ ~ 60 ns). The enhanced coherence times closely approach the spin-lattice relaxation time, which is the theoretical limit[50]. We observe that DD has no impact on the strong modulation observed in the spin-echo $T_2$ measurement. This modulation remains unchanged in both amplitude and frequency as $N$ increases. We also note that the finite duration of the MW pulses sets the shortest pulse sequence time $t_s$. Moreover, particularly for large $N$, we expect contributions from pulse errors and $T_1$ relaxation[44,45]. Please refer to Supplementary Fig. 2 for a better visualization of the data related to the experiments with $N = 300, 600, 800,$ and 1000 π-pulses.

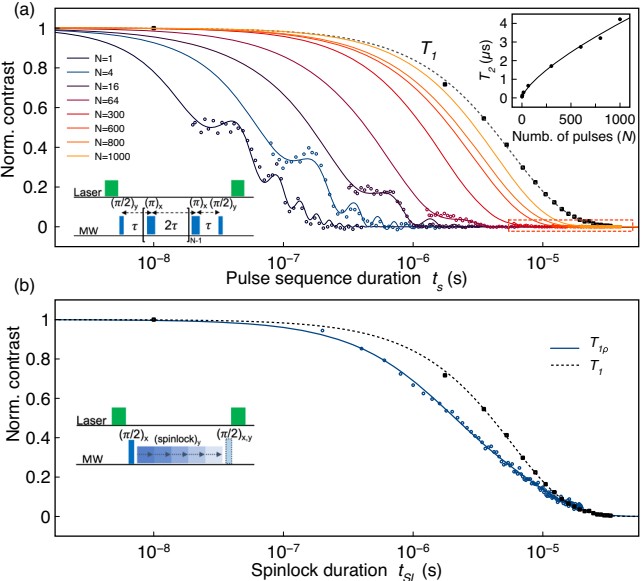

**Fig. 2 | Extension of coherence for $V_B^-$ in hBN. a** Measured decoherence curves (semi-log plot) for the $V_B^-$ spins under the effect of different CPMG decoupling pulse sequences with an increasing number of π-pulses $N$. Colored dots and solid lines are data points and fits, respectively. In black, the $T_1$ measurement of Fig. 1g is shown with the corresponding fit as dotted line. Bottom, left: dynamical decoupling (CPMG) pulse sequence. Bottom, right: the data enclosed in the dotted box are shown in Supplementary Fig. 2. Inset: plot of the coherence times vs the number of π-pulses $N$ utilized in each CPMG experiment. The data points fit a power function $f(N) = a \times N^s$ where $s$ ~ 0.71. **b** Rotating-frame spin-lattice relaxation time ($T_{1\rho}$) measured with the depicted spinlock sequence (blue), in comparison with the $T_1$ measurement (black).

In the upper-right inset of Fig. 2a, we show the increase of the coherence times versus the number of $\pi$-pulses $N$. The plot depicts a sub-linear dependence that can be fit with a simple power function $f(N) = a \times N^s$ where $s \sim 0.71$, which is in good agreement with the theoretical dependence of $T_2 \propto N^{2/3}$ expected for a Lorentzian spin bath, in the limit of long spin bath correlation times $t_c \gg \tau$[39,48]. These results demonstrate that dynamical decoupling works effectively for our sample, similar to the situation encountered for nitrogen-vacancy (NV) ensembles in diamond in the presence of high-density paramagnetic impurities (50-100 ppm) and natural abundance $^{13}$C nuclear spins[39,45] or for $V_{Si}^-$ in 4H-SiC[51].

We also note that a factor of two increase in coherence time by a CPMG protocol was shown by conventional electron paramagnetic resonance (EPR) spectroscopy, although under very different experimental conditions (magnetic field of ~3 T and cryogenic temperature of ~50 K)[20].

An alternative approach that preserves coherence relies on generating $V_B^-$ dressed spin states using spinlock pulse sequences[52,53]. After optical initialization of the defects in the $|0\rangle$ state, a $(\pi/2)_x$-pulse generates their coherent superposition which is then locked along the $y$-axis of the Bloch sphere by a spinlock pulse. We measure the spin-lattice relaxation time in the electron-spin rotating-frame ($T_{1\rho}$) with the experiment depicted in Fig. 2b. Here, the $V_B^-$ fluorescence contrast is monitored while increasing the spinlock pulse duration. We demonstrate that the $V_B^-$ dressed-states and, therefore, the electronic spin coherence can survive ~130× longer than in the case of the spin-echo $T_2$ (Fig. 1h). The decay of the spinlocked coherence can be fit by a bi-exponential function (see Methods) and reaches the ~$T_1$ relaxation time.

## Sensing RF signals with $V_B^-$ defects in hBN

With the improved coherence time, the application of advanced quantum sensing protocols is now possible. In particular, we explore RF sensing by alternatively using pulsed dynamical decoupling (pDD) techniques, or continuous dynamical decoupling (cDD) schemes. In both cases, we test the response of the $V_B^-$-based sensor to an RF magnetic field of the form $B_{RF}(t) = b_{RF}\cos(2\pi\nu_{RF}t + \phi_{RF})$. Here, $b_{RF}$, $\nu_{RF}$ and $\phi_{RF}$ are the RF amplitude, frequency and phase, respectively. The RF field is aligned perpendicular to the hBN flake's plane (i.e., quantization $z$-axis of the spin defects) and its phase is unlocked with respect to the control pulse sequence. A more detailed explanation on quantum sensing of RF fields with hBN is provided in the Supplementary Note 3.

In pDD sequences, the $V_B^-$ spins are brought into a superpostion state through a $\pi/2$-pulse and the RF field induces a relative phase $\theta_{pDD}$ to the $V_B^-$ electronic spin's superposition, such as: $|\psi\rangle = (|0\rangle \pm e^{i\theta_{pDD}(t_s)}|1\rangle)/\sqrt{2}$. For detection, the signal is rectified by applying a train of $N$ equidistant $\pi$-pulses at times $\tau = 1/(4\nu_{RF})$. This allows the phase to be accumulated over an interrogation time of $t_s = 2N\tau$, reaching a maximum value of $\theta_{pDD}(t_s) = (2/\pi)\gamma b_{RF}t_s$, where $\gamma$ is the electron gyromagnetic ratio. This accumulated phase is then converted to a spin population difference by a final $\pi/2$-pulse and read out optically. This scheme acts as a pass-band filter with center frequency $f_c = \frac{1}{4\tau}$ and bandwidth $\Delta f = \frac{1}{t_s}$. This means that the protocol enhances the effect of RF fields oscillating within the bandwidth $\Delta f$ while suppressing all other field fluctuations. Supplementary Note 3 presents experimental evidence that the $V_B^-$ spin ensembles are subject to the sensing mechanism described above. As a first example, we use the pulsed dynamical decoupling sequence (XY8-$N$), depicted in Fig. 3a, to sense the RF frequency $\nu_{RF}$. The XY8-$N$ protocol is similar to the CPMG sequence described earlier, but it employs a distinct pattern of MW-pulse phases that alters the spin rotation axis at each $\pi$-pulse. This variation is designed to minimize the impact of pulse errors. We have chosen to utilize the XY8-$N$ sequence instead of CPMG as it typically exhibits superior performance in detecting RF fields[54–56] (see Supplementary Note 4). In Fig. 3b, we employ a sequence with 16 $\pi$-pulses

(XY8-2) and test the sensor's response by sweeping $\tau$ while keeping $\nu_{RF}$ at a defined value. Fluorescence contrast dips appear at the expected $\tau$ for two different values of $\nu_{RF}$ and their lineshapes can be fitted by the expected sinc-squared function[3], as detailed in the Methods part. In Supplementary Note 6, we explore the frequency range that can be probed using the XY8-$N$ protocols on our sample, which spans from ~10 MHz to 40 MHz. In addition, we provide an analysis of the criteria for optimizing the pDD sequence and determining the optimal number of pulses for a specific sensing frequency. These experiments directly demonstrate the sensor's ability to detect an unknown RF signal. In Fig. 3c, we show the dependence of the detected signal on the number of $\pi$-pulses. As anticipated, the XY8-2 dip narrows down with respect to the one obtained with the XY8-1 sequence, however, simultaneously, the signal-to-noise ratio (SNR) decreases due to decoherence. More details about the experimentally probed spectral bandwidth can be found in Section 4 of Supplementary Note 3.

An alternative approach for sensing RF signals are cDD pulse sequences which are based on rotating-frame magnetometry[57–59] (Fig. 3d). They exploit the $V_B^-$ coherence locked on the transversal plane by a spinlock pulse of duration $t_{SL}$ and variable amplitude $\Omega$. Matching the $V_B^-$ Rabi frequency $\nu_R$ with the sensing frequency $\nu_{RF}$, such that: $\Omega/(2\pi) = \nu_R = \nu_{RF}$ drives transitions between $V_B^-$ dressed-states $|\pm\rangle = (|0\rangle \pm ie^{i\theta_{SL}(t_{SL})}|1\rangle)/\sqrt{2}$ and induces a relative phase $\theta_{SL}(t_{SL}) = \frac{1}{2}\gamma b_{RF}t_{SL}$. As in the previous pDD method, this accumulated phase is translated to a spin population difference by a final $\pi/2$-pulse and optically read out, providing direct information about the strength of the RF magnetic field. The spinlock method can also be seen as a pass-band filter whose center frequency is given by the spinlock amplitude $f_c = \nu_R$, and whose bandwidth $\Delta f$ is set by the spinlock time $t_{SL}$ (see Section 7 of Supplementary Notes 3 for a more detailed description of the spinlock experiment). Figure 3e illustrates RF sensing using the spinlock pulse sequence. We apply a sample RF field to the $V_B^-$ sensor and sweep the spinlock MW amplitude $\Omega/(2\pi)$ in a range of a few tens of MHz around the RF frequency $\nu_{RF}$. Similar to what was obtained with the XY8-2 protocol in Fig. 3b, we observe dips at the matching conditions: $\nu_{RF} = \Omega/(2\pi)$. Moreover, we explore the possible range of detectable RF frequencies which, in this case, is limited by the spinlock amplitude $\Omega$. As shown in Supplementary Fig. 16, the spinlock sequence gives us access to RF frequencies in the range of $2\,MHz \lesssim \nu_{RF} \lesssim 25\,MHz$. In Fig. 3f, we investigate the sensor's response by probing it with various spinlock durations $t_{SL}$. We observe an increase in SNR and a reduction in linewidth until reaching an optimal duration of approximately 0.5 μs. Beyond this point, we notice a gradual degradation of the signal and broadening of the dips, approaching the limits imposed by the $T_{1\rho}$ and $T_1$ relaxation times. A discussion of these aspects can be found in Section 7 of Supplementary Note 3.

Additionally, in Supplementary Note 7, we demonstrate coherent control of the $V_B^-$ dressed-states and probe their evolution during the spinlock pulse[57,59,60]. The detection of such states can be particularly useful in scenarios where pseudo-Zeeman spin states are preferred over bare-states due to the convenience of modulating their energy splitting through the microwave amplitude. Applications that benefit from these states include nuclear-spin hyperpolarization, which can enhance the sensitivity of magnetic resonance spectroscopy, enable the use of nuclear spins as quantum registers, or improve the accuracy of quantum simulators[12,61,62]. A system like the $V_B^-$ in hBN has the potential to benefit all of these areas. As evident from the experimental results, advanced quantum sensing protocols can be applied using spin defects in 2D materials and for qubit-control in 2D quantum technologies[26,46,63–65].

## Sensing of RF signals with arbitrary frequency resolution

The frequency resolution in dynamical decoupling sequences is typically restricted by the duration of the pulse sequence and thus by the coherence time. This is a particular limitation for $V_B^-$ defects in

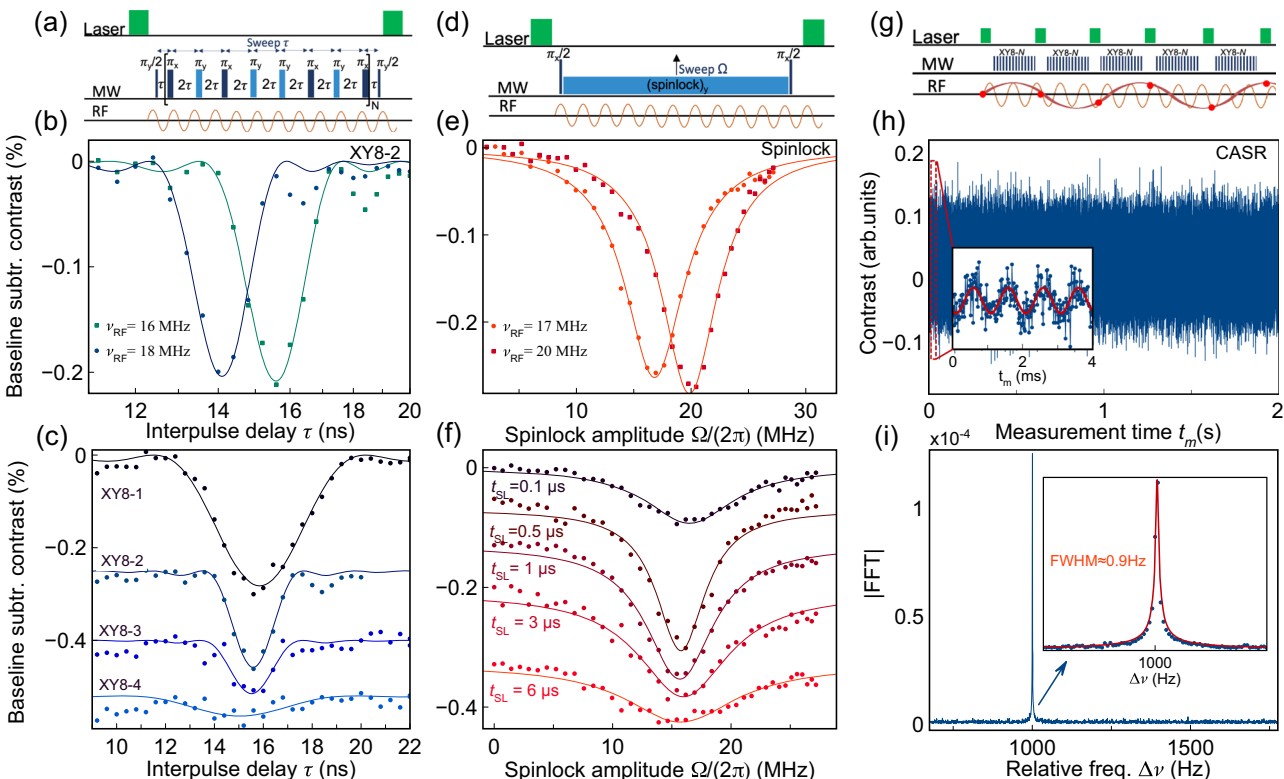

**Fig. 3 | RF-sensing with $V_B^-$ in hBN. a** Pulsed-DD protocol used for sensing RF signals. An XY8-2 pulse sequence was utilized for the experiments in **b**, **c** setting a test RF field at a constant frequency $\nu_{RF}$ and sweeping $\tau$. **b** Response of the $V_B^-$ sensor as fluorescence contrast dips occurring at matching of the interpulse delay $\tau$ with the RF signal period according to: $\tau = 1/(4\nu_{RF})$. The data points (dots) are best fitted with a (sinc)$^2$ function. **c** Dependence of the lineshape on the number of pulses for XY8-1,2,3,4 sequences setting $\nu_{RF} = 16$ MHz. **d** Spinlock-based protocol for RF-sensing. A test RF field was set at a constant frequency $\nu_{RF}$ and the spinlock pulse duration was kept fixed ($t_{SL} = 0.5 \mu s$) while its amplitude $\Omega$ was swept.

**e** Spectral response as fluorescence contrast dips occurring at matching of the spinlock amplitude with the RF frequency according to $\nu_R = \nu_{RF}$. **f** Dependence of the lineshape on the spinlock duration. **g** Coherently Averaged Synchronized Readout (CASR) protocol. XY8-2 subsequences are synchronized to the sensing frequency for 2 seconds. **h** Time domain CASR signal. Each detected point corresponds to an optical readout represented by the red circles in **g**. In the inset, a zoom-in of the signal in the first 4ms shows the oscillations at $\Delta\nu = 1000$ Hz. **i** Fourier transformation of the signal in **h** yields a sharp peak at the relative frequency $\Delta\nu$. A Lorentzian linefit of the peak (red line) results in a 0.9 Hz linewidth.

hBN, but it can be overcome by applying sensing schemes that integrate classical heterodyne detection with quantum sensing, which allow spectral resolutions that are independent on the properties of the qubit[66-68]. In particular, we demonstrate a sensing scheme called coherently averaged synchronized readout (CASR)[66]. As explained in detail in Section 8 of Supplementary Note 3, the fundamental idea of the CASR protocol is the synchronization of a series of pDD sequences (e.g., XY8-$N$) with the sensing RF magnetic field which is kept phase-locked to the control sequence. For a slight detuning of the RF frequency $\nu_{RF}$ from the pulse sequence synchronization frequency $\nu_{pDD} = 1/(4\tau)$, the fluorescence oscillates in time domain at the difference frequency $\Delta\nu = \nu_{pDD} - \nu_{RF}$. This synchronization technique enables arbitrary frequency resolution since it is no longer limited by the intrinsic coherence time of the solid-state spin system, but rather by the clock stability of the experimental setup responsible for the reiteration of the pulse sequences for an arbitrarily long time.

Figure 3g illustrates the pulse scheme utilized for our experiments, which consists of a train of XY8-2 sequences, which are synchronized with an RF signal of frequency $\nu_{RF} = 18$ MHz. We run the sequence for a total measurement time $t_m = 2$ s (Fig. 3h), resulting in a time-dependent measurement that can be Fourier transformed to give a peak with sub-Hertz linewidth (Fig. 3i). Importantly, this method can render our ultra-thin quantum sensor capable of sensing oscillatory magnetic fields with a high frequency resolution, with possible applications in nanoscale magnetic resonance spectroscopy[67,68]. Finally, we used the CASR protocol to experimentally determine the sensitivity of

our $V_B^-$ detector which resulted to be ~2–3 $\mu$T Hz$^{-1/2}$. Details about the sensitivity measurement can be found in Section 8 of Supplementary Note 3.

## Comparison with state-of-the-art NV-diamond sensors
To draw a parallel with other competing systems, we examined and compared the RF sensing capabilities of our $V_B^-$ sensor with other state-of-the-art quantum sensors based on ensembles of NV centers in diamond[11,69,70]. We explain the details of this comparison in the Supplementary Note 5. In a simplified picture, the sensitivity of optically-active spin-based sensors depends on the number of detected photons ($n$) and on the spin coherence times ($T_2$) according to: $\eta \propto \frac{1}{\sqrt{n}} \frac{1}{\sqrt{T_2}}$[11,69,71].

Considering the shorter coherence times (on the order of nanoseconds for our $V_B^-$ sample versus microseconds timescales for NV-sensors) and the lower quantum efficiency (~0.03% vs. ~70%)[72], we anticipate that our $V_B^-$ sensor will exhibit lower sensitivity compared to the NV-diamond sensors. Indeed, assuming similar experimental conditions, such as the same light collection efficiency, our hBN sample is expected to give a sensitivity on the order of a few $\mu$T Hz$^{-1/2}$, consistent with the experimental findings. In contrast, ensembles of NV centers in diamond typically provide sensitivities on the order of nT Hz$^{-1/2}$ [69,73]. While this may initially be perceived as a limitation, it is important to acknowledge the significant signal enhancements that $V_B^-$ sensors can provide in various applications, particularly in nanoscale spin sensing. These enhancements arise from the 2D properties of the host material, which enable an ideal interface with target materials, reduced

presence of dangling bonds and surface imperfections, and optimal proximity between the sensors and the target spins for effective interaction. Initial experiments, such as those shown in the recent work of Durand et al.[74] investigating $V_B^-$ functionality in few-layer hBN substrates, are crucial steps towards revealing the full potential of these systems.

In conclusion, this work addresses the short spin coherence of the hBN spin defects, a severe limitation for their application in quantum technology. Utilizing pDD protocols, we achieve a ~70 times extension of the coherence with respect to the single spin-echo $T_2$, approaching the limit of the longitudinal relaxation time $T_1$. Furthermore, by generating $V_B^-$ dressed-states with spinlock pulses, we demonstrate an extension of the coherence up to ~7.5 μs, overcoming the spin-echo $T_2$ by more than two orders of magnitude. The improved coherence times enable us to demonstrate RF signal detection in several complementary experiments. In particular, we demonstrate that dynamical decoupling protocols, such as XY8-$N$ or spinlock pulse sequences, are functional for sensing RF signals. Furthermore, we show that despite the intrinsically short $V_B^-$ coherence, sensing RF frequencies with high (sub-Hz) frequency resolution is also possible using quantum heterodyne detection approaches, such as CASR or Qdyne schemes[66–68,75]. While the experiments indicate that our $V_B^-$ sensor is less sensitive to RF fields compared to state-of-the-art NV-diamond quantum sensors, the distinctive potential of hBN as a Van der Waals material to form intimate interfaces with target samples can possibly mitigate this limitation. Additionally, the experiments reported, along with the possible miniaturization and integration of these sensors into 2D heterostructures, set the stage for the establishment of nanoscale spin sensing for the exploration of emergent phenomena in low-dimensional quantum materials and devices. Finally, the improved spin control and the intriguing nuclear-spin environment surrounding the defects open a promising path to the realization of multi-qubit registers for quantum sensors and quantum networks integrated into ultra-thin structures.

## Methods

### Experimental setup

Initialization of the $V_B^-$ ensemble is realized with a 532 nm laser (Opus 532, Novanta photonics) at a power of ~100 mW (CW). Laser pulses are timed by an acousto-optic modulator (3250-220, Gooch and Housego) with typical pulse durations of 5 μs. The laser light is reflected by a dichroic mirror (DMLP650, Thorlabs) after which it is focused on the hBN flake by an objective (CFI Plan Apochromat VC 20×, NIKON) with a numerical aperture of NA = 0.75. Photoluminescence (PL) is collected by the same objective and alternatively focused by a tube lens on: (1) an avalanche photodiode (APD) (A-Cube-S3000-10, Laser Components) for the spectroscopic path; (2) a camera (a2A3840-45ucBAS, Basler) for imaging the sample. The excitation green light and possible unwanted fluorescence from other defects are filtered out using a long-pass filter with a cut-on wavelength of 736 nm (Brightline 736/128, Semrock). The output voltage of the APD is digitized with a data acquisition unit (USB-6221 DAQ, National Instruments). An arbitrary waveform generator (AWG) up to 2.5 GS/s (AT-AWG-GS2500, Active Technology) is used to synchronize the experiment and generate rectangular arbitrary-phased RF pulses for $V_B^-$ spin control. For up-conversion of the AWG RF frequency (typically 250 MHz) to the MW frequency required for $V_B^-$ driving, mixing with a local oscillator generator (SG384, Stanford Research Systems) is realized by means of an IQ mixer (MMIQ0218LXPC 2030, Marki). The amplified microwave pulses (ZHL-16W-43-S+, Mini-Circuits) are delivered by a gold stripline to the hBN sample. A permanent magnet underneath the sample holder is utilized for applying the magnetic field of ~8 mT. The ODMR frequency is used to determine the magnetic field strength $B_0$ as well as the $V_{B\,[0,-1]}^-$ resonance frequency $f_{V_B^-}$. The radio wave signals used to

demonstrate RF magnetometry are produced by an RF waveform generator (DG1022z, Rigol) connected to a 30 W amplifier (LZY-22+, Mini-Circuits). A small wire loop placed in the proximity of the sample was used for the RF delivery.

### Sample preparation

The hexagonal boron nitride van der Waals flakes were cleaved and tape-exfoliated starting from hBN seed crystals (2D semiconductors) on a silicon wafer with a 70 nm top oxide layer. We then implanted the exfoliated samples at the ion beam facility (Helmholtz-Zentrum Dresden-Rossendorf, HZDR) with a helium ion fluence of $3 \times 10^{14}$ ions/cm$^2$ at an energy of 3 keV. Once the photoluminescence spectrum of the boron vacancies was verified, we transferred the boron vacancy containing hexagonal boron nitride with a standard dry transfer method on top of a gold stripline evaporated on a sapphire substrate. The gold stripline was connected to a printed circuit board using several bond-wires in parallel to improve impedance matching and enabling high power MW delivery for short Rabi pulses.

### ODMR measurement

For all experiments throughout this work, 5 μs-long laser pulses were used for initialization/readout. The ODMR measurement displayed in Fig. 1e) was performed using a 1 μs-long MW pulse at ~1mW power for the driving of the $V_B^-$ spin populations. The fluorescence contrast was monitored while increasing the MW frequency. For normalization and noise cancellation purposes, a second reference sequence was applied right after the first one, differing only from the MW being off[76]. The single data points result from dividing the fluorescence readouts of the two sequences. 10,000 averages for each data point and further 600 averages of the full frequency sweep were recorded. After baseline subtraction, the lineshape was fitted with seven Gaussian functions of the form $a[\exp(-\ln(2)(f - f_0)^2/\mathrm{LW}^2)]$ whose amplitude $a$ and frequency $f_0$ were used as fitting parameters whereas the half-width-half-maximum LW was kept at a constant value of 22 MHz. The relevant fit parameters are reported in Supplementary Table 4.

### Rabi experiment

Rabi oscillations were recorded by setting the MW frequency in the center of the ODMR spectrum and sweeping the MW-pulse duration $t_p$ in stepwise increments. Different MW power (37, 33, 28 and 23 dBm) were used corresponding to the different datasets in Fig. 1f). For each point, 10,000 averages were acquired following the same normalization and noise cancellation procedure used for the ODMR experiments. In Fig. 1f, the Rabi oscillations were fitted using the following function: $1 - c/2 + c/2(\cos(2\pi\nu_R t + \Phi))[a \times \exp(-t/T_a) + b \times \exp(-t/T_b)]$, where $c$ is a term for coherence normalization, $\nu_R$ the Rabi frequency, $\Phi$ a phase term and the bi-exponential factor accounts for the damping of the oscillation. Likely due to the dephasing processes causing damping and out-of-step effects of the Rabi oscillation, we observed a significant discrepancy ($\gtrsim 20\%$) between the Rabi frequencies obtained directly from the fits and the expected frequencies based on the first minimum of the Rabi oscillation curve that corresponds to the actual spin population inversion, as determined by $\theta_{\mathrm{flip}} = 2\pi\nu_R t_p$, where $t_p$ represents the duration of the pulse. To address this issue, we calculated the first derivatives of the Rabi curves, fitted them using a polynomial spline, and identified the $x$-intercept as the precise minimum position, indicative of the π-pulse duration. Consequently, for each Rabi experiment, we measured π-pulse durations of $(7.5 \pm 0.1)$, $(10.1 \pm 0.2)$, $(15.3 \pm 0.3)$, and $(28 \pm 1)$ ns. Subsequently, we calculated the corresponding Rabi frequencies $\nu_R$ using the aforementioned flip-angle formula: $\nu_R = 1/(2t_\pi)$. To illustrate the relationship between the Rabi frequencies and the microwave power, we plotted the $\nu_R$ values against the square root of the microwave power. Performing a linear fit ($y = ax$), we obtained a slope of $a = (32 \pm 2)$ MHz/$\sqrt{\mathrm{W}}$ (Fig. 1f, inset).

## $T_1$ measurement

The $T_1$ time (see Fig. 1g) was measured by sweeping the time $\tau$ between initialization and readout pulses in a range from 10 ns to ~30 μs. For noise cancellation purposes, a second reference sequence was applied right after the first one, differing only for a MW $\pi$-pulse inserted after the first laser pulse. The data points were then obtained by dividing the readouts of the two consecutive sequences. Every point was averaged 100,000 times and the whole time sweep was averaged four times. The $T_1$ decay curve was fitted with a simple mono-exponential function of the form $a(\exp(-\tau/T_1))$ where $T_1 = (5.84 \pm 0.05)$ μs.

## $T_2$ measurement

The spin-echo sequence was set up following the scheme $[(\pi/2)_y - \tau - (\pi)_x - \tau - (\pi/2)_y]$ where $\tau$ is swept from 10 ns to ~200 ns. Referencing for noise cancellation was achieved by alternating the last MW-pulse of the spin-echo sequence from $\pi/2$ to $(3/2)\pi$[76]. Every point was averaged 100,000 times and the whole time sweep was averaged five times. The decay was fitted with the function: $a \times e^{-(2\tau/T_2)^c} + b \times \cos(2\pi f \tau) \times e^{-(\tau/T_f)}$ where the first stretched exponential yields the time constant $T_2 = (58.5 \pm 0.4)$ ns with $c = 1.03 \pm 0.01$. To guide the eye in the figures, we fitted the oscillation superimposed on the decay with a cosine function with $f = 44.5$ MHz, multiplied by an exponential term which reproduces the damping of the observed oscillation with $T_f = 71$ ns. The relevant fit parameters are reported in Supplementary Table 5.

## CPMG experiments

We applied the same scheme utilized for the $T_2$ measurement, that is monitoring the spin-echo signal while increasing the free evolution time $\tau$. Multiple decoherence curves were acquired for pulse sequences with growing number $N$ of $\pi$-pulses, up to $N = 1000$. The data were normalized according to the following procedure. First, for all datasets, the time axis was multiplied by a scaling factor $S = 2N$ accounting for the real time elapsed between the first and the last $\pi/2$-pulse. Then, the decay curve corresponding to the single $\pi$-pulse experiment (spin-echo) was fitted by the function: $a \times e^{-(t_s/T_2)^c} + b \times \cos(2\pi f t_s/S) \times e^{-(t_s/T_f)}$. The point of maximum contrast, namely maximum intensity of the spin-echo, when no decoherence has occurred yet, has been extrapolated from the fit in correspondence to the time $t = 0$ of the decay. Then, the contrast of all datasets has been normalized to this value and then fitted with the same function, keeping the amplitudes $a$ and $b$ locked. The modulation frequency $f$ has been used as a fitting parameter only for the datasets with $N = 1$, 4, and 16, yielding $f = (44.1 \pm 0.5)$, $(44.7 \pm 0.6)$, $(45.9 \pm 0.8)$ MHz, respectively. For the other datsets, $f$ was kept locked to 44.5 MHz. The fitted $T_2$ values and relative exponents $c$ are reported in Supplementary Table 6. The dependence of the coherence times on the number of pulses shown in the inset of Fig. 2a was fitted with a simple power function $f(N) = a \times N^s$ where $a = (29 \pm 1)$ ns and $s = 0.71 \pm 0.05$.

## $T_{1\rho}$ measurement

$T_{1\rho}$ was measured using a pulse sequence $[(\pi/2)_x - d - (\text{spinlock})_y - d - (\pi/2)_x]$ and monitoring the resulting fluorescence contrast while applying step-by-step increments of the spinlock pulse duration $t_{SL}$. 3.5 ns-long $\pi/2$-pulses were used and the delay times $d$ kept to a value as short as possible (~1–2 ns), in order to avoid significant dephasing during this time. The spinlock amplitude was kept at 10% of the MW amplitude utilized for the $\pi/2$-pulses. 100,000 averages were taken for each data point. The resulting curve fits a bi-exponential decay of the form: $a(\exp(-t_{SL}/T_{1a})) + b(\exp(-t_{SL}/T_{1b}))$, with $a = 0.48 \pm 0.02$, $T_{1a} = (1.38 \pm 0.09)$ μs, $b = 0.52 \pm 0.02$, $T_{1b} = (7.52 \pm 0.21)$ μs.

## XY8-$N$ experiments

RF sensing was performed by means of a XY8-2 sequence following the scheme: $(\pi/2)_y[[-\tau - (\pi)_\phi - \tau]_8]_2$, with the following $\pi$-pulses phase-scheme: $\phi = [x - y - x - y - y - x - y - x]_2$. $\pi/2$ and $\pi$-pulse durations of

respectively 4 ns and 8 ns have been used. An RF field was applied by a wire loop in the vicinity of the sample and the RF frequency, $\nu_{RF}$, was held at a constant value. The interpulse duration, $\tau$, was varied between 10 and 20 nanoseconds, which is in a range where the sensing frequency was expected to match the condition $\nu_{RF} = \frac{1}{4\tau}$. The RF phase was left unlocked with respect to the pulse sequence, resulting in random phase variations for each repetition of the pulse sequence. Referencing for noise cancellation was achieved by alternating the last MW-pulse of the spin-echo sequence from $\pi/2$ to $(3/2)\pi$. Every point was averaged 100,000 times and the whole sweep averaged 49 times. The observed dips were first baseline-corrected by subtracting a dataset of the same experiment performed with no RF signal and then fitted with the following function: $a/2[\sin(2\pi\nu_{RF}N(\tau - \tau_0))/(2\pi\nu_{RF}N(\tau - \tau_0))]^2$. In the fittings, we set $\nu_{RF}$ at a constant sample frequency and leave $a$, $\tau$ and $N$ as free parameters. The raw data before baseline subtraction are shown in Supplementary Note 9. The same procedure has been applied for the data in Fig. 3c. All relevant fit parameters are reported in Supplementary Tables 7 and 8.

## Spinlock protocol

For sensing, the same spinlock sequence described above for the measurement of the $T_{1\rho}$ was utilized. A $\pi/2$-pulse of 4 ns and a spinlock duration $t_{SL} = 0.5$ μs were used. The RF phase was left unlocked to the pulse sequence, resulting in random phase variations for each repetition of the pulse sequence. The RF frequency, $\nu_{RF}$, was held at a constant value and the spinlock amplitude $\Omega$ was swept, corresponding to sweeping the Rabi frequency $\nu_R$ until matching the condition $\nu_R = \nu_{RF}$. The observed dips were first baseline-corrected by subtracting a dataset of the same experiment performed with no RF signal and then fitted with Lorentzian functions of the type: $a/(1 + (\nu - \nu_{RF})^2/LW^2)$. The raw data before baseline subtraction are shown in Supplementary Note 9. The same procedure was applied for the data in Fig. 3f. All relevant fit parameters are reported in Supplementary Tables 9 and 10. Since the spinlock amplitude is swept by varying the peak-to-peak voltage (V.p.p.) of our AWG, a calibration procedure was applied to report the spinlock amplitude in frequency units (MHz) (See inset of Supplementary Fig. 16).

## CASR protocol

We applied a pulse sequence which was synchronized with the sensing RF signal and consisted of concatenated XY8-2 subsequences repeated so that the timing between them was an integer number of the RF period. For each XY8 sequence, we used 5 μs-long laser pulses and $\pi$-pulses of 8 ns for initialization/readout and MW manipulation, respectively. We sensed an RF frequency $\nu_{RF}$ ~ 18 MHz by setting the interpulse delay $\tau = 14$ ns. To get a relative frequency $\Delta\nu = 1000$ Hz, the RF was shifted to $\nu_{RF} = 18.001$ MHz. A total measurement time $t_m$ of 2 seconds was utilized. The detected time trace was then Fourier transformed and the magnitude plotted in Fig. 3i. The resulting peak in the frequency domain was fitted with a modified Lorentzian function to determine the full width half-maximum. The signal was the result of 1000 averages.

## Data availability

Source data are provided with this paper[77]. The Source Data used in this study are available in the ZENODO database under accession code [https://zenodo.org/record/8135158]. All other data that support the findings of this study are available from the corresponding author upon reasonable request. Source data are provided with this paper.

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

## Acknowledgements

R.R. and D.B.B. would like to thank Dr. Steffen Glaser for stimulating discussions. R.R. thanks Nick Neuling for his help in the laboratory. This study was funded by the Deutsche Forschungsgemeinschaft (DFG, German Research Foundation)—412351169 within the Emmy Noether program. R.R. acknowledges support from the DFG Walter Benjamin Program (Project RI 3319/1-1) and the PNRR-PE "National Quantum Science and Technology Institute" (NQSTI), NextGenerationEU 2022-RTDA-4445. G.S. acknowledges the Marco Polo program of University of Bologna (grant 2022) for funding her stay abroad. J.J.F. and D.B.B. acknowledge support from the DFG under Germany's Excellence Strategy-EXC 2089/1-390776260 and the EXC-2111 390814868 as well as by the Bayerisches Staatsministerium für Wissenschaft und Kunst through project IQSense via the Munich Quantum Valley (MQV). Support by the Ion Beam Center (IBC) at HZDR is gratefully acknowledged.

## Author contributions

R.R., M.S., and D.B.B. conceived the idea of RF sensing with $V_B^-$ in hBN. R.R. and D.B.B. designed the research. R.R. carried out the experiments and the simulations. M.S., C.Q., and P.J. prepared the sample and fabricated the microstructure for MW delivery. R.R. M.S. and S.M. built the ODMR setup. J.C.H. and J.P.L. helped in the optimization of the experimental setup, J.C.H. contributed to the theoretical derivations. F.B. programmed the pulse sequences. G.S. contributed to the experiments for sensitivity estimations. G.V.A., U.K., and M.H. were responsible for the ion implantation for the generation of the $V_B^-$ centers in the hBN substrate. A.V.S. and J.J.F. advised on several aspects of theory and experiments. R.R., M.S., and D.B.B. analyzed the data. R.R., J.C.H., and D.B.B. wrote and reviewed the manuscript with inputs from all authors.

## Funding

## Competing interests

All authors declare no competing interests.
