## [Peer Review File · Nature Communications]

REVIEWER COMMENTS

Reviewer #1 (Remarks to the Author):

The boron vacancy center in hBN has recently emerged as a promising spin qubit/quantum sensor. As with other potential spin qubits, improvement of coherence times of VB⁻ is a key research goal in the development of practical quantum computing and quantum sensing solutions. One technique for extending coherence times is the use of dynamical decoupling schemes, such as the CPMG and spinlock sequences. In the manuscript by Rizzato et al., CPMG and spinlock have been shown to significantly enhance coherence times of VB⁻. To the best of my knowledge, the application of CPMG to VB⁻ was only reported once, in Ref. [20]. But there, the experiments were carried out only at cryogenic conditions (T=50K). Thus, the room-temperature results reported in the current manuscript represent a major advancement. Last but not least, the demonstrated applicability of the VB⁻ center for AC magnetometry paves the way for the use of this defect for nanoscale NMR spectroscopy (as an alternative to the NV center in diamond). Overall, I think this is a well-executed and valuable contribution to the field of quantum technologies and solid-state physics. Therefore, it is well-suited in terms of content and impact for Nature Communications.

I have only a few minor suggestions that could further improve the manuscript:

1. I suggest providing a more detailed discussion of the T₁ (a.k.a. spin-lattice) relaxation time of VB⁻. As stated in the manuscript, T₁ defines the upper limit of the coherence time and thus is an important physical parameter. However, the way T₁ is discussed in the present version of the manuscript leaves the impression that its value is taken for granted. For their sample, the authors measure T₁ to be about 6 μs, but, for example, in Ref. [18], a value of 18 μs is reported. What could be the reason for this difference? Can T₁ depend on sample preparation? Are there ways to increase T₁?

2. In Section 5, the authors showed that the VB⁻ center can detect an alternating magnetic field with a frequency of ~1 MHz using the CASR protocol. They concluded that this proves the applicability of VB⁻ for nanoscale NMR. But, as far as I understand, even in the case of the diamond NV center, a very long total measurement time is required to achieve an acceptable resolution and signal-to-noise ratio when using CASR to detect the Larmor-precession induced AC magnetic fields in real samples. Won't this become a prohibitive limitation for the VB⁻ center, which has inferior coherent properties compared to NV? I think a brief discussion and/or some estimates will help reinforce the conclusions and provide the reader with a more complete picture.

3. Lines 114-115: I find the sentence about the expected effect of fast decoherence on the electron spin of VB- a bit confusing. This sentence follows from the observation of the inhomogeneous broadening of the VB- electron-spin transitions. In this context, wouldn't it be more accurate to state that hyperfine interactions affect the electron spins of VB-, which in turn can cause its fast decoherence?

Reviewer #2 (Remarks to the Author):

This is a report on ODMR of boron vacancies in hBN. The main results are the extension of the coherence time from a spin echo time of ~ 100 ns to a few μ s using a pulsed decoupling scheme. The method is adapted to measure an rf signal applied by a second antenna.

A few comments

1. Recently, Ramsay et al [Nature Comm. 2023 14 461] reported extension of coherence time from ~ 100 ns to μ s time using a continuous dynamic decoupling method. The methods used are different, but the main outcome is similar.

2. In this work, various pulsed schemes are used to measure the coherence decay, and give rise to different time-parameters that carry different information on the spin-environment coupling. It is very difficult to follow a discussion where everything is labelled as T_2 .

3. In fig 1(f), the caption refers to four traces, there are only three. In the inset, why is the intercept not zero? What is the B-field applied? The exponent of the spin-echo measurement should be reported for completeness.

4. In fig. 2(a) for N-large the maximum contrast is limited by the duration of the pulse sequence. If $T_2^N(\text{CPMG}) \sim N^{2/3}$, and the time duration of the sequence scales as N, at some point you start to lose. It may help to show the time cost in fig 2(b). It would be useful to resize fig 2(c) to be on the same scale as fig 2(a) to allow a more direct assessment of the spin-lock vs CPMG method. Also the x-axis labels could be more descriptive than "time". What are the implications of this data for sensor performance?

In sections 4 and 5 “we demonstrate ...set of quantum sensing protocols..” For me, this is potentially the most important part of the paper, since this is where the novelty lies. But this section needs a lot of improvement.

5. Three different sensing protocols are presented. In each case, precisely what does the sensor respond to? What quantity is the sensor being used to measure? For example, what component of the Brf is detected? Does the sensor respond to the power or one of the quadratures of the field at the selected frequency? What determines the bandwidth of the sensor? Are the DD and rf fields synchronized? In other words, what is it that the device does? This functionality needs to be clearly stated. What experiments are needed to demonstrate that functionality?

6. Is there a demonstration of a sensing protocol? Fig3(b,e,c,f) are all calibration measurements, where the rf field gives the control parameter. There is no demonstration of a measurement of an unknown parameter of brf.

7. It looks like the bare minimum of data is presented. For example, in fig 3(a) there are 3 different values of tau presented over a range of 2-3 linewidths. Where is the graph in the supplement or main text quantifying the accuracy of the relationship? When there are statements like “Due to short coherence time T_2 , the pulsed DD fails...frequencies $\omega < 1/T_2$ ”, where is the data supporting that statement? These are room temperature measurements, on an ensemble, where mechanical and photo stability of the setup should not be an issue. An automated measurement can run overnight and at the weekends, and there is a supplement.

8. In sec. 5 a lot is made of the sub-hertz linewidth. Is it a big deal? Surely, this is just Nyquist theorem?

9. What is it that makes a good sensor? If it takes 2000s to measure an rf frequency synchronized with the sensor drive, is that useful for magnetic resonance spectroscopy? Is this competitive with rival defect systems?

Overall, this is an album of spin control experiments. It is not surprising that a protocol that works in diamond also works to some extent in hBN. In my evaluation of this work, I am asking what do I learn about the applications potential of VB- in hBN? At a “high” level the applications potential of hBN is an important question, and I agree that this is a good research direction. Section 3 reports a valuable contribution to the field. But in the current state, sections 4 and 5 do not belong in an academic journal, for the reasons outlined in 5-9. This is fixable. Mostly, the discussion needs to be more thoughtful. The experiments need to be geared to addressing a well defined, and meaningful

set of research questions that make a meaningful, appraisal of the hBN device capabilities, and this probably will require further measurements.

Reviewer #3 (Remarks to the Author):

The manuscript reported an experimental study of extending the coherence time of VB- spin defects in hBN. Spin defects in hBN are promising spin qubit systems, but the short spin coherence time limits their applications in quantum technology. This study applies dynamical decoupling techniques to suppress magnetic noise and extend the spin coherence time by nearly two orders of magnitude, which comparable to the results of the recently reported continuous concatenated dynamic decoupling method [Nat. Commun. 14, 461(2023)]. In addition, this study also demonstrates VB-defects as quantum sensors to detect RF signals.

I consider that this study shows good applicability of dynamical decoupling techniques for VB-defects, which is timely and significant for the applications of spin defects in hBN. However, the paper also has some shortcomings, which should be revised before the publication.

1. In Figure 2a, we find that the modulation gets weaker for larger N, why the authors stated that dynamical decoupling does not affect the strong modulation in the spin-echo T₂ measurement in the second paragraph of page 5? Whether the modulation frequency is dependent on N?

2. The authors show the results of extending the coherence time of VB- defects using CPMG pulse sequences and spinlock pulse sequences, and we note that they also use XY8-N pulse sequences in part 4. I suggest the authors show the results of extending the coherence time of VB- defects using XY8-N pulse sequences.

3. The authors show sensing RF signals with VB- defects in part 4, and the technologies used are advanced and the results are very good. However, we note the authors only kept fixed and sweep the RF frequency. In the practical application, the RF frequency is a physical quantity to be measured, so how to measure a fixed RF frequency? I suggest that the author would better add a demonstration of practical measurements.

4. In Figure 3c, the authors show that the spectral width narrows down and the signal-to-noise ratio (SNR) decreases by increasing N. Is there a criterion that picks out a best N to balance the relationship between spectral width and SNR? It would be helpful for practical application.

5. Why can we observe hyperfine structure in Figure 3(e), but not in figure 3(f)?

We also have more minor questions and comments about the manuscript.

6. The authors stated that “Spin defects in hexagonal Boron Nitride (hBN) attract increasing interest for quantum technology since they represent optically-addressable qubits in a van der Waals material.” in abstract. Note that not all spin defects are optically-addressable, and this needs to be modified.

7. In the second paragraph of part 4, the authors stated that “The DD sequence then acts as a narrow-band RF filter and the VB- superposition accumulates a maximal phase $\theta(t_s) = (2/\pi)\gamma_{bRF} t_s$...leading to a dip in the fluorescence intensity.” The authors should specify the meaning of γ .

8. There are two articles on extending the coherence time of VB- defects recently [Nat. Commun. 14, 461(2023), arXiv preprint arXiv: 2210.11485 (2022)], I suggest the authors could cite them.

9. In the first paragraph of page 7, the authors stated that “In addition, we study the spectral response of the sensor with different spinlock durations t_{SL} (Figure 3e)”, Figure 3e should be Figure 3f.

REVIEWER COMMENTS

Reviewer #1 (Remarks to the Author):

The boron vacancy center in hBN has recently emerged as a promising spin qubit/quantum sensor. As with other potential spin qubits, improvement of coherence times of VB- is a key research goal in the development of practical quantum computing and quantum sensing solutions. One technique for extending coherence times is the use of dynamical decoupling schemes, such as the CPMG and spinlock sequences. In the manuscript by Rizzato et al., CPMG and spinlock have been shown to significantly enhance coherence times of VB-. To the best of my knowledge, the application of CPMG to VB- was only reported once, in Ref. [20]. But there, the experiments were carried out only at cryogenic conditions (T=50K). Thus, the room-temperature results reported in the current manuscript represent a major advancement. Last but not least, the demonstrated applicability of the VB- center for AC magnetometry paves the way for the use of this defect for nanoscale NMR spectroscopy (as an alternative to the NV center in diamond). Overall, I think this is a well-executed and valuable contribution to the field of quantum technologies and solid-state physics. Therefore, it is well-suited in terms of content and impact for Nature Communications.

I have only a few minor suggestions that could further improve the manuscript:

1. I suggest providing a more detailed discussion of the T₁ (a.k.a. spin-lattice) relaxation time of VB-. As stated in the manuscript, T₁ defines the upper limit of the coherence time and thus is an important physical parameter. However, the way T₁ is discussed in the present version of the manuscript leaves the impression that its value is taken for granted. For their sample, the authors measure T₁ to be about 6 μs, but, for example, in Ref. [18], a value of 18 μs is reported. What could be the reason for this difference? Can T₁ depend on sample preparation? Are there ways to increase T₁?

We are grateful to the reviewer for bringing up this important point. It is possible that the value of T₁ varies from sample to sample due to uncontrollable amounts of paramagnetic impurities in the material. This is supported by the fact that there is some variability in the reported T₁ measurements in the literature. For example, in our study, we measured a T₁ of approximately 6 μs after performing helium-ion implantation with an energy of 3 keV and a dose of 3×10^{14} ions cm⁻². In contrast, the T₁ value of 18 μs reported in the study cited by the reviewer [18] was obtained using a different sample preparation method involving neutron irradiation. Furthermore, Baber *et al.* [19] found a T₁ time of 10 μs, which is more similar to our result. This has been obtained with an implantation procedure involving carbon-ion implantation with an energy of 10 keV and a similar dose to ours of 1×10^{14} ions cm⁻². Our use of a smaller implantation energy might have led to the formation of defects closer to the surface that might be even richer in paramagnetic impurities, which could explain our shorter T₁ value compared to that of Baber et al. [19]. However, our T₁ time seems consistent with the data shown in Gong et al., arXiv preprint: 2210.11485 2022, where they used a sample preparation very similar to ours (please check FIG.1 (e) in their study).

A comment about this has been added at lines 153-157 of the resubmitted main text.

2. In Section 5, the authors showed that the VB⁻ center can detect an alternating magnetic field with a frequency of ~1 MHz using the CASR protocol.

Likely this is a typo; in section 5, we demonstrated the sensing of an 18 MHz RF signal.

They concluded that this proves the applicability of VB⁻ for nanoscale NMR. But, as far as I understand, even in the case of the diamond NV center, a very long total measurement time is required to achieve an acceptable resolution and signal-to-noise ratio when using CASR to detect the Larmor-precession induced AC magnetic fields in real samples. Won't this become a prohibitive limitation for the VB⁻ center, which has inferior coherent properties compared to NV? I think a brief discussion and/or some estimates will help reinforce the conclusions and provide the reader with a more complete picture.

The Reviewer is correct: the coherence time sets a limitation to both the resolution and sensitivity of the sensing methods and this is even more dramatic for VB⁻ in hBN where the coherence times are short compared to other systems (NV centers). In fact, the experiments demonstrated in Section 4 (XY8-N sequences and spinlock), represent the fundamental building blocks for the development of more advanced AC sensing protocols and they all suffer from this limitation. However, the key message of our work is that, experiments like CASR (or Qdyne) have been introduced exactly to overcome such a physical limitation and are functional also with VB⁻ in hBN. We agree with the Reviewer that this point is very important and deserves a deeper analysis and discussion. For this reason, we performed new experiments and estimated the sensitivity of our V_B⁻ sensor. This is included in the new Supplementary Notes 3 (Section 8) and 5. We estimate a sensitivity of approx. 2 μT/sqrt(Hz). This can be considered as a large value compared to what is typically achieved with NV centers in diamond (~ nT/sqrt(Hz)). However, we have to consider that:

- 1) it is achieved in a material that has never been optimized for better coherence/relaxation times/luminescence properties;
- 2) using hBN as a sensor has the advantage of being able to position it much closer to the target compared to other bulk spin defects like NV centers in diamond. For example, hBN defects may be positioned at around 1 nm from the sample compared to the typical 5 nm for NV centers. This can result in a significant increase in signal (up to ~5⁶), which goes up to the sixth power of the distance between the sensor and the target.

We also made an effort to compare our VB⁻ sensor with state-of-the-art quantum sensors that rely on ensembles of NV-centers in diamond. We have included this comparison in a newly written Discussion section in the main text. Additional details and information can be found in Supplementary Note 5.

3. Lines 114-115: I find the sentence about the expected effect of fast decoherence on the electron spin of VB⁻ a bit confusing. This sentence follows from the observation of the inhomogeneous broadening of the VB⁻ electron-spin transitions. In this context, wouldn't it be more accurate to state that hyperfine interactions affect the electron spins of VB⁻, which in turn can cause its fast decoherence?

Thank you for pointing out this passage that was indeed unclear. We have rephrased it accordingly at lines 133-136.

Reviewer #2 (Remarks to the Author):

This is a report on ODMR of boron vacancies in hBN. The main results are the extension of the coherence time from a spin echo time of ~ 100 ns to a few μ s using a pulsed decoupling scheme. The method is adapted to measure an rf signal applied by a second antenna.

A few comments

1. Recently, Ramsay et al [Nature Comm. 2023 14 461] reported extension of coherence time from ~ 100 ns to μ s time using a continuous dynamic decoupling method. The methods used are different, but the main outcome is similar.

The submission of our manuscript and this publication overlapped; therefore, it was not included. We have now added the citation of this work and mentioned it in the main text at lines 88-91. We would like to highlight that our research is fundamentally different and provides new information on the potential use of VB-spin defects in hBN for sensing RF fields. To achieve this, we extended the coherence time using dynamical decoupling methods.

2. In this work, various pulsed schemes are used to measure the coherence decay, and give rise to different time-parameters that carry different information on the spin-environment coupling. It is very difficult to follow a discussion where everything is labelled as T_2 . We added a superscript indicating the number of π -pulses utilized (ex. $T_2^{(1000)}$). Furthermore, in the resubmitted text, we avoided using the term " T_2 " to refer to the coherence time, except when specifically discussing the spin-echo coherence time achieved with a single π -pulse.

3. In fig 1(f), the caption refers to four traces, there are only three.

We report all four traces now.

In the inset, why is the intercept not zero?

We are very grateful to the Reviewer for bringing this to our attention. We acknowledge that there was an error in the plot, and we have now corrected the inset. Additionally, we have realized that in the previous version, we labeled the MW powers with the values prior to amplification. We have now rectified and updated the labeling accordingly.

What is the B-field applied?

All experiments have been performed at a bias field of 8 mT. The information has been added to the figure's caption.

The exponent of the spin-echo measurement should be reported for completeness.

We are not entirely certain what the Reviewer is referring to regarding the exponent. If they're inquiring about the exponential function used to fit the experimental data, we have documented this in detail in the Methods part at lines 532-544.

4. In fig. 2(a) for N -large the maximum contrast is limited by the duration of the pulse sequence. If $T_2^N(\text{CPMG}) \sim N^{(2/3)}$, and the time duration of the sequence scales as N , at some point you start to lose. It may help to show the time cost in fig 2(b).

We thank the Reviewer for this comment, which raises a point that indeed deserved a more detailed discussion in the manuscript. In effect, for CPMG-like pulsed schemes, the sensitivity of the method can be enhanced by using N π -pulses that have the effect of extending the T_2 time. However, the number of π -pulses used increases the measurement time linearly,

whereas the T_2 time increases sub-linearly. As a result, for a given RF frequency to sense, an optimal number of pulses can be identified for the best outcome.

We added a new chapter in the Supplementary Information (Supplementary Note 6) where, supported by calculations, we show how the sensitivity of the sensing scheme depends on the duration of the pulse sequence and at the same time on the coherence properties which are enhanced by the dynamical decoupling protocol.

It would be useful to resize fig 2(c) to be on the same scale as fig2(a) to allow a more direct assessment of the spin-lock vs CPMG method.

Fig.2 has been updated to show both experiments, CPMG (Fig. 2a) and Spinlock (Fig. 2b), on the same x-axis scale for an easier comparison. Additionally, the T_1 measurement curve is now included in both figures to provide a visual reference for the performance of the two different dynamical decoupling methods in extending the coherence times.

Also the x-axis labels could be more descriptive than “time”.

The x-axis labels have been fixed accordingly.

What are the implications of this data for sensor performance?

The data reported in Fig. 2a have to be read together with the discussion presented in the previous answer. Here, the effect of the number of pulses and thus on the coherence time extension is analyzed considering the simultaneous increase of the overall measurement time. This allows us to identify the optimal conditions where the hBN device presented in the work can be utilized as a sensor of RF fields.

In sections 4 and 5 “we demonstrate ...set of quantum sensing protocols..” For me, this is potentially the most important part of the paper, since this is where the novelty lies. But this section needs a lot of improvement.

We have thoroughly reconsidered and reworked sections 4 and 5 of the main text. To address the Reviewer's request for more information and description without making the text too long, we also have prepared an extensive Supplementary Information that provides many details regarding the presented methods.

5. Three different sensing protocols are presented. In each case, precisely what does the sensor respond to?

What quantity is the sensor being used to measure? For example, what component of the B_{rf} is detected?

Does the sensor respond to the power or one of the quadratures of the field at the selected frequency?

What determines the bandwidth of the sensor?

Are the DD and rf fields synchronized?

In other words, what is it that the device does? This functionality needs to be clearly stated.

What experiments are needed to demonstrate that functionality?

To clarify all these points, we included a comprehensive chapter in the Supplementary Information (Supplementary Note 3). Here, we provide a summary of the theory behind RF sensing with spin defects with a specific focus on V_B^- centers in hBN, and highlights our research findings regarding this system.

6. Is there a demonstration of a sensing protocol? Fig3(b,e,c,f) are all calibration measurements, where the rf field gives the control parameter. There is no demonstration of a measurement of an unknown parameter of b_{rf} .

The experiments shown in Fig. 3 involved sweeping the RF frequency while keeping the inter pulse delay τ constant, until the sensor's expected response was triggered (e.g., a fluorescence contrast dip when matching condition $t=1/(4\nu_{RF})$ was met). While these experiments demonstrate the basic sensor's functionality, they could be considered as "calibration" experiments as they did not showcase the sensor's ability to respond to unknown parameters, such as an unknown frequency of the RF signal.

To address this limitation, we conducted new experiments that replaced the previously reported ones in Fig. 3 b, c, e. In this new set of experiments, we kept the RF test signal at a fixed frequency (e.g., 16 and 18 MHz) and varied the spin interrogation time (τ) for the XY8 protocol, and the spinlock amplitude Ω for the spinlock sequence.

These experiments demonstrate the sensor's ability to respond to unknown parameters and provide a more comprehensive understanding of its sensing capabilities.

New data is shown in the revised (b), (c), (e) and (f) subfigures of Fig. 3:

7. It looks like the bare minimum of data is presented. For example, in fig 3(a) there are 3 different values of tau presented over a range of 2-3 linewidths.

Where is the graph in the supplement or main text quantifying the accuracy of the relationship?

All information about the data fittings is in the Methods part of the main text, Section 8.3 Pulse sequences, normalizations and fittings. This was updated and modified for more clarity and details. In particular, all fitting procedures have been better described and the relevant fit parameters have been reported in the Supplementary Note 8.

We thank the Reviewer for this comment. Paying more attention to fitting uncertainties, we found a small bug in the code (a bracket misplacement) used to fit the coherence data. This caused the stretched exponential function, which was used to analyze the coherence curves, to be improperly quantified together with the associated uncertainties. As a consequence, some values reported in the first version of the manuscript have changed, and in particular: the coherence times, the stretch exponents (as in the tables below) and the exponent of the dependence of the coherence times on the number of pulses $f(N) = a \times N^s$ from $s = 0.62$ to $s = 0.71$. These variations are all marked in red in the resubmitted manuscript. For instance, the table previously reported in the Methods part:

N (π -pulses)	T_2 [ns]	c
1	90 \pm 2	1.425 \pm 0.026
4	220 \pm 4	1.403 \pm 0.025
16	500 \pm 5	1.435 \pm 0.016
64	940 \pm 11	1.299 \pm 0.015
300	1759 \pm 32	1.309 \pm 0.024
600	2776 \pm 35	1.265 \pm 0.016
800	3434 \pm 30	1.257 \pm 0.011
1000	4204 \pm 44	1.310 \pm 0.014

Now is reported in the Supplementary Note 8 and reads:

TABLE S6. Fitted coherence time constants and corresponding stretch-exponents c for the data shown in Figures 2 (a). Parameters with * have been kept locked in fitting.

N (π -pulses)	T_2 [ns]	c
1	63 \pm 1	1.04 \pm 0.03
4	138 \pm 3	0.98 \pm 0.04
16	294 \pm 9	0.88 \pm 0.04
64	653 \pm 13	0.97*
300	1710 \pm 370	1.12 \pm 0.17
600	2730 \pm 249	1.02 \pm 0.06
800	3209 \pm 265	1.00 \pm 0.05
1000	4220 \pm 493	1.06 \pm 0.01

We apologize for that. However, this issue did not affect the main story of the paper in any way.

When there are statements like “Due to short coherence time T_2 , the pulsed DD fails...frequencies nurf $<1/T_2$ ”, where is the data supporting that statement?

We performed new experiments with the aim of exploring the sensing frequency range that can be probed using the current protocols. The results are reported in Fig. S13 of Supplementary Note 6. The XY8-N sequence is limited by the coherence time at the low end, making it challenging to measure frequencies below 10 MHz. At the high end, we were able to measure a maximum frequency of more than 40 MHz, the limit being set by the maximum microwave (MW) power, which determines the minimal duration of the MW pulses.

Similarly, using the spinlock protocol, we found challenging to sense RF frequencies below 5 MHz, mainly dictated by the inefficiency of the spinlock pulse to keep the spin “locked” when set at a low MW amplitude. The upper limit was again determined by the maximum MW amplitude that could be matched with the sample RF. We avoided to test the spinlock sequence at full MW power to not damage the sample. These data about the spinlock experiments are shown in Figure S16 of Supplementary Note 9.

These are room temperature measurements, on an ensemble, where mechanical and photo stability of the setup should not be an issue. An automated measurement can run overnight and at the weekends, and there is a supplement.

8. In sec. 5 a lot is made of the sub-hertz linewidth. Is it a big deal? Surely, this is just Nyquist theorem?

We are not totally sure if we understood this point correctly (regarding the Nyquist theorem?). Perhaps the reviewer means that there's nothing particularly noteworthy about a sensor response with a sub-hertz linewidth signal, since this is simply a consequence of Fourier transforming a time-domain signal of a few seconds. This is true, in fact, the key is not the Fourier transformation itself but the possibility, even in a system with short coherence times like V_B^- in hBN, to detect an RF signal in principle for arbitrarily-long times. In our work the CASR (or Qdyne) methods are applied for the first time to V_B^- defects in hBN and this constitutes the basis to perform high-resolution NMR spectroscopy at the nanoscale. We inserted a more detailed description of the CASR experiment in the Section 8 of Supplementary Note 3.

9. What is it that makes a good sensor? If it takes 2000s to measure an rf frequency synchronized with the sensor drive, is that useful for magnetic resonance spectroscopy? Is this competitive with rival defect systems?

We acknowledge the Reviewer's comment that this point is crucial and warrants further analysis and discussion.

For this reason, we performed new experiments and estimated the sensitivity of our V_B^- sensor. This is included in the Section 8 of Supplementary Note 3. We estimate a sensitivity of $\sim 2 \mu\text{T}/\sqrt{\text{Hz}}$. This can be considered as a large number compared to what is typically achieved with NV centers in diamond ($\sim \text{nT}/\sqrt{\text{Hz}}$). However, we have to consider that:

- 1) it is achieved in a material that has never been optimized for better coherence/relaxation times/luminescence properties;
- 2) using hBN as a sensor has the advantage of being able to position it much closer to the target compared to other bulk spin defects like NV centers in diamond. For example, an hBN spin defect can be positioned at 1 nm from the sample compared to 5 nm for NV centers. This can result in a significant increase in signal (up to $\sim 5^6$), which goes up to the sixth power of the distance between the sensor and the target.

We also made an effort to compare our V_B^- sensor with state-of-the-art quantum sensors that rely on ensembles of NV-centers in diamond. We have included this comparison in a newly written Discussion section in the main text. Additional details and information can be found in the Supplementary Note 5.

Overall, this is an album of spin control experiments. It is not surprising that a protocol that works in diamond also works to some extent in hBN.

hBN is a fundamentally different material than diamond as each nucleus in hBN carries spin, and the V_B^- center is surrounded by three very strongly coupled nitrogen nuclei. This is in contrast to diamond's NV-centers, where the material is essentially spin-free, except for the occasional presence of ^{13}C , and the nitrogen nucleus, whose coupling to the electron spin is much weaker.

Based on these considerations, we were initially pretty skeptical that such a system could be used as a sensor for oscillating magnetic fields, given the expected magnetic noise. We were pleasantly surprised to find that V_B^- defects in hBN not only worked as a sensor but did so

without any special engineering or material optimization, such as isotopic enrichments or optimized implantation conditions. We are truly impressed by these results.

In my evaluation of this work, I am asking what do I learn about the applications potential of VB- in hBN?

At a “high” level the applications potential of hBN is an important question, and I agree that this is a good research direction. Section 3 reports a valuable contribution to the field. But in the current state, sections 4 and 5 do not belong in an academic journal, for the reasons outlined in 5-9. This is fixable.

Mostly, the discussion needs to be more thoughtful. The experiments need to be geared to addressing a well defined, and meaningful set of research questions that make a meaningful, appraisal of the hBN device capabilities, and this probably will require further measurements. We would like to express our sincere gratitude to the reviewer for providing insightful comments that have significantly enhanced the quality of our manuscript.

We made our best to improve the manuscript following the Reviewer’s guidelines:

- 1) we conducted new experiments and replaced the data presented in Figure 3;
- 2) we conducted new experiments to quantify the sensitivity achieved with our sample;
- 3) we made new calculations of the expected sensitivity based on the theory established with NV-centers in diamond. Then, we wrote a new Discussion section where we compare the performance of our sensor with the ones of other NV-diamond based sensors;
- 4) We conducted new experiments probing the range of RF frequencies that can be detected with our working sensor;
- 5) We made new calculations to indicate how to optimize the pulse sequences depending on the certain frequency range of interest;
- 6) We wrote an extensive Supplementary Information describing the basics of the sensing methods;
- 7) We reported all fitting details;
- 8) We rewrote section 4 and 5 that the Reviewer found particularly problematic.

We hope the reviewer will appreciate the improvements we have made based on her/his valuable input.

Reviewer #3 (Remarks to the Author):

The manuscript reported an experimental study of extending the coherence time of VB- spin defects in hBN. Spin defects in hBN are promising spin qubit systems, but the short spin coherence time limits their applications in quantum technology. This study applies dynamical decoupling techniques to suppress magnetic noise and extend the spin coherence time by nearly two orders of magnitude, which comparable to the results of the recently reported continuous concatenated dynamic decoupling method [Nat. Commun. 14, 461(2023)]. In addition, this study also demonstrates VB- defects as quantum sensors to detect RF signals.

I consider that this study shows good applicability of dynamical decoupling techniques for VB- defects, which is timely and significant for the applications of spin defects in hBN. However, the paper also has some shortcomings, which should be revised before the publication.

1. In Figure 2a, we find that the modulation gets weaker for larger N, why the authors stated that dynamical decoupling does not affect the strong modulation in the spin-echo T2 measurement in the second paragraph of page 5? Whether the modulation frequency is dependent on N?

The frequency of the modulation observed in the decoherence curves shown in Fig2a is constant and independent on the number of pulses N. The decrease in frequency observed in fig.2a has its origin in the scaling factor that is applied to the x-axis to account for the actual pulse sequence duration. In fact, when the scaling is present, each curve can be fitted with the equation reported in Section 8.3.2 of the Methods part:

$$a(\exp(-t_s/T_2))^c + b(\exp(-t_s/T_f))\cos(2\pi f t_s/S)$$

Here, $S = 2N$ is the scaling factor and $t_s = S\tau = 2N\tau$.

Thus, for each curve with increasing N, the modulation frequency f is constant.

In the following, the decoherence curves (same dataset in Fig. 2a for $N=1,4,16$), are shown without scaling the x-axis, indicating that the frequency is constant.

2. The authors show the results of extending the coherence time of VB- defects using CPMG pulse sequences and spinlock pulse sequences, and we note that they also use XY8-N pulse sequences in part 4. I suggest the authors show the results of extending the coherence time of VB- defects using XY8-N pulse sequences.

We thank the Reviewer for this comment which raises up an interesting point.

To better examine this aspect, we performed new experiments and in particular we probed the extension of the coherence time by means of an XY8-N pulse sequence (shown in the Supplementary Note 4). These data show that, with XY8-N, a dramatic drop in contrast occurs after ~ 100 pulses (vs. 1000 with CPMG). The XY8-N sequence still extends the V_B^- coherence time, however showing worse performance in preserving the prepared spin state with respect to the CPMG experiment.

On the other hand, the CPMG showed to be less sensitive than the XY8-N since it did not give any appreciable signal upon application of the RF fields in the same condition utilized for the XY8-2 experiments in Figure 3b of the main text (see the revised Figure 3 below). The reason for this is that the CPMG technique is a pulsed spin-lock protocol that can efficiently preserve the spin polarization along the axis in which the initial state was prepared. However, it has the side effect of eliminating all components that are perpendicular to it.

In contrast, the XY-protocol is not as efficient at "locking" the spin state onto a specific axis, but it can compensate for pulse errors and better preserve an arbitrary spin state that may have developed perpendicular components due to phase accumulation. A comment about this aspect has been added in the main text at lines 262-267 and in the Supplementary Note 4.

3. The authors show sensing RF signals with VB- defects in part 4, and the technologies used are advanced and the results are very good. However, we note the authors only kept fixed and sweep the RF frequency. In the practical application, the RF frequency is a physical quantity to be measured, so how to measure a fixed RF frequency? I suggest that the author would better add a demonstration of practical measurements.

The experiments shown in Fig. 3 involved sweeping the RF frequency while keeping the interpulse delay τ constant, until the sensor's expected response was triggered (e.g., a fluorescence contrast dip when matching condition $t=1/(4\nu_{RF})$ was met). While these experiments demonstrated the sensor's functionality, they could be considered as "calibration" experiments as they did not showcase the sensor's ability to respond to unknown parameters, such as an unknown frequency of the RF signal.

To address this limitation, we conducted new experiments that replaced all the ones previously reported in Fig. 3 b, c, e, and f. In this new set of experiments, we kept the RF test signal at a fixed frequency (e.g., 16 and 18 MHz) and varied the spin interrogation time (τ) for the XY8 protocol, whereas the spinlock amplitude Ω for the spinlock sequence.

These experiments directly demonstrate the sensor's ability to respond to unknown parameters and provide a more comprehensive understanding of its sensing capabilities.

New data is shown in the revised (b), (c), (e) and (f) subfigures of Fig. 3:

4. In Figure 3c, the authors show that the spectral width narrows down and the signal-to-noise ratio (SNR) decreases by increasing N . Is there a criterion that picks out a best N to balance the relationship between spectral width and SNR? It would be helpful for practical application. We appreciate the Reviewer for her/his valuable comment. Indeed, it is possible to select the best conditions for sensing a given RF frequency (or a range of frequencies). To explain this process, we have included a new chapter in the Supplementary Information (Supplementary Note 6). This chapter includes calculations that show how the sensitivity of the sensing scheme is influenced by the duration of the pulse sequence, as well as the coherence properties that are improved by the dynamical decoupling protocol. Based on these considerations, the optimal number of pulses can be determined according to the coherence properties of our sensor.

5. Why can we observe hyperfine structure in Figure 3(e), but not in figure 3(f)?

We apologize for any confusion that may have arisen. The Lorentzian fits under the dip in Figure 3(e) are not a direct signature of the hyperfine structure, but rather they were intended to represent the inhomogeneously broadened nature of the spinlock signal dips. We previously analyzed this feature extensively in a prior paper from our group (Rizzato *et al.*, Phys. Rev. Applied 17, 024067(2022)), but we have since decided that it is not necessary to discuss it here as it may add further confusion. We have replaced all figures related to the spinlock experiments with the data from new experiments.

We also have more minor questions and comments about the manuscript.

6. The authors stated that “Spin defects in hexagonal Boron Nitride (hBN) attract increasing interest for quantum technology since they represent optically-addressable qubits in a van der Waals material.” in abstract. Note that not all spin defects are optically-addressable, and this needs to be modified.

Thanks, we amended the abstract accordingly.

7. In the second paragraph of part 4, the authors stated that “The DD sequence then acts as a narrow-band RF filter and the VB- superposition accumulates a maximal phase $\theta(t_s) = (2/\pi)\gamma_{\text{RF}} t_s$...leading to a dip in the fluorescence intensity.” The authors should specify the meaning of γ .

We added the meaning of γ at line 254.

8. There are two articles on extending the coherence time of VB- defects recently [Nat. Commun. 14, 461(2023), arXiv preprint arXiv: 2210.11485 (2022)], I suggest the authors could cite them.

We have now included the citation of these works and added a mention of them in the main text. The first at lines: 88-92, the second at line 153:

9. In the first paragraph of page 7, the authors stated that “In addition, we study the spectral response of the sensor with different spinlock durations t_{SL} (Figure 3e)”, Figure 3e should be Figure 3f.

We thank the Reviewer for bringing the inconsistency to our attention. We amended it accordingly.

REVIEWERS' COMMENTS

Reviewer #1 (Remarks to the Author):

In the revised manuscript, the authors have thoroughly addressed my comments and recommendations. Furthermore, the changes prompted by the suggestions from other reviewers have significantly enhanced the clarity, comprehensiveness, and generality of the data presentation, as well as the robustness of the conclusions. Specifically, the authors have added the results of new sensing experiments that provide a more comprehensive understanding of the capabilities and limitation of the VB- center. In addition, the expanded discussion of the data has made the manuscript more accessible and reader-friendly, particularly for non-expert readers. As a result, I believe that the manuscript has undergone substantial improvements and is now suitable for publication.

Reviewer #2 (Remarks to the Author):

The second part of the manuscript has improved a lot. Sec. 4 now contains a sensing experiment. Measurements, analysis and discussion of the sensor performance, including an estimate of sensitivity are now reported. Previously, this was another ODMR spectroscopy paper that justified its significance by talking the talk on the importance of hBN for B-field sensing. It is now a paper that walks the walk. It enquires into, and informs the discussion on the merits of hBN for B-field sensing, and represents genuine progress on the question. Hence, I support publication.

A few minor comments.

1. (L244) What component/direction of the rf B-field is applied? Is the signal field synchronized with the control field? In other words, a summary of precisely what the sensor responds to would help the customer select the scheme for their application.
2. In sec. 6, expression for sensitivity needs to be supported by reference, as does statement in L389 on NV center sensitivities. It may be good to elaborate on whether the sensitivities quoted are for rf-fields or use the same CASR method.

3. I did not express myself clearly, on point 7. In for example fig. 3(d), the authors argue that the minimum of the dip occurs when the spin-lock frequency matches the signal frequency, allowing one to measure the signal frequency. This is true in theory, and confirmed for only two frequencies spaced by less than a linewidth. In practice, one wants to know how close to ideal the relationship is, and over what range the sensor can be used.

I do not expect the authors to address every possible aspect of the sensor capabilities in a single report. I raise the issue to clarify the point only. The authors have done enough to placate me.

Reviewer #3 (Remarks to the Author):

The authors have answered my questions and revised the manuscript to improve its quality. Thus I recommend it to be published in Nature Communications.

REVIEWERS' COMMENTS

Reviewer #1 (Remarks to the Author):

In the revised manuscript, the authors have thoroughly addressed my comments and recommendations. Furthermore, the changes prompted by the suggestions from other reviewers have significantly enhanced the clarity, comprehensiveness, and generality of the data presentation, as well as the robustness of the conclusions. Specifically, the authors have added the results of new sensing experiments that provide a more comprehensive understanding of the capabilities and limitation of the VB- center. In addition, the expanded discussion of the data has made the manuscript more accessible and reader-friendly, particularly for non-expert readers. As a result, I believe that the manuscript has undergone substantial improvements and is now suitable for publication.

Reviewer #2 (Remarks to the Author):

The second part of the manuscript has improved a lot. Sec. 4 now contains a sensing experiment. Measurements, analysis and discussion of the sensor performance, including an estimate of sensitivity are now reported. Previously, this was another ODMR spectroscopy paper that justified its significance by talking the talk on the importance of hBN for B-field sensing. It is now a paper that walks the walk. It enquires into, and informs the discussion on the merits of hBN for B-field sensing, and represents genuine progress on the question. Hence, I support publication.

A few minor comments.

1. (L244) What component/direction of the rf B-field is applied? Is the signal field synchronized with the control field? In other words, a summary of precisely what the sensor responds to would help the customer select the scheme for their application.

We would like to thank once again the Reviewer for her/his great assistance in improving the manuscript. Initially, we chose not to include these details in the main text to maintain readability. However, we have now added the information regarding the phase-synchronization and the direction of the RF B-field at lines 234-240 and at line 327.

Furthermore, we provided all the relevant details in Supplementary Note 3.

Specifically:

- the direction of the RF B-field is mentioned in lines 119-125.
- In Section 5 of Supplementary Note 3, "Slope and Variance Detection," at lines 257-263, we specify which protocols are performed with the signal field synchronized with the control field (slope detection) and which ones are conducted without such synchronization (variance detection).

2. In sec. 6, expression for sensitivity needs to be supported by reference, as does statement in L389 on NV center sensitivities. It may be good to elaborate on whether the sensitivities quoted are for rf-fields or use the same CASR method.

We have addressed the concern by including the references for the sensitivity expression at line 354, as well as at line 362 for NV center sensitivities. The sensitivities mentioned are specifically for RF fields and are based on the detection of coherent RF signals using slope detection. To clarify this point further, at lines 512 to 528 in Supplementary Note 5, we have explicitly stated that all our arguments regarding sensitivity rely on slope detection of coherent RF signals. Furthermore, we have added a mention of the case of variance detection of incoherent signals to provide a more comprehensive understanding of the topic.

3. I did not express myself clearly, on point 7. In for example fig. 3(d), the authors argue that the minimum of the dip occurs when the spin-lock frequency matches the signal frequency, allowing one to measure the signal frequency. This is true in theory, and confirmed for only two frequencies spaced by less than a linewidth. In practice, one wants to know how close to ideal the relationship is, and over what range the sensor can be used.

The Referee's point can be clarified by referring to the set of experiments illustrated in Suppl. Figure 13 (a) for the pulsed dynamical decoupling sequences and in Suppl. Figure 16 for the spinlock experiments.

For instance, in the latter case (reported below), the relationship the Referee would like to see is clearly displayed. The lefthand figure shows the raw data prior to baseline subtraction. In each of these experiments, we set a different RF frequency (5 to 25 MHz) and the spinlock amplitude is gradually increased using our Arbitrary Waveform Generator (AWG). The spinlock amplitude values are reported in units of Volt peak-to-peak (V_{pp}), since it directly corresponds to the readings obtained from our device.

For each RF frequency, we observe a dip centered at a defined spinlock amplitude value. This series of experiments enables us to calibrate our sensor, and the inset graph depicts the relationship between the sensing RF and the corresponding matching spinlock amplitude.

Regarding the usable range of the sensor, this is determined by the available spinlock amplitude, which is limited by the microwave (MW) power provided by our equipment. In principle, a range of ~5-50 MHz can be achieved with our setup with both protocols (pDD and cDD). However, as indicated in the righthand figure below, where the typical growing baseline has been subtracted (and as mentioned in the comment on lines 608-617), we observed line broadening at 25 MHz, likely due to MW heating. Consequently, we decided to discontinue probing higher frequencies to avoid any potential damage to our sample.

I do not expect the authors to address every possible aspect of the sensor capabilities in a single report. I raise the issue to clarify the point only. The authors have done enough to placate me.

Reviewer #3 (Remarks to the Author):

The authors have answered my questions and revised the manuscript to improve its quality. Thus I recommend it to be published in Nature Communications.